# Pt/Pd Decorate MOFs Derived Co-N-C Materials as High-Performance Catalysts for Oxygen Reduction Reaction

Yuhang Jiang, Dejing Zhu, Xiangchuan Zhao, Zhaoyun Chu, Liping Zhang, Yue Cao *[ID] and Weimeng Si *

School of Material Science and Engineering, Shandong University of Technology, Zibo 255000, China; j17864306182@163.com (Y.J.); 18753364007@163.com (D.Z.); 17864301031@163.com (X.Z.); czy1921381183@163.com (Z.C.); zlp1998225@163.com (L.Z.)
* Correspondence: cao-yue@foxmail.com (Y.C.); siweimeng@foxmail.com (W.S.)

**Abstract:** We report here, a strategy to prepare Pt/Pd nanoparticles decorated with Co-N-C materials, where Co-N-C was obtained via pyrolysis of ZIF-67 directly. As-prepared Pt/Pd/Co-N-C catalysts showed excellent ORR performance, offered with a higher limit current density (6.6 mA cm$^{-2}$) and similar half-wave potential positive ($E_{1/2}$ = 0.84 V) compared with commercial Pt/C. In addition to an ORR activity, it also exhibits robust durability. The current density of Pt/Pd/Co-N-C decreased by only 9% after adding methanol, and a 10% current density loss was obtained after continuous testing at 36,000 s.

**Keywords:** oxygen reduction reaction; PtPd nanoparticles; ZIF-67; Co-N-C

## 1. Introduction

Oxygen reduction reaction (ORR) is an important cathode reaction in the fuel cell, and the activity of ORR catalysts greatly affects the performance and conversion efficiency of the fuel cell [1–3]. At present, Pt has been used the most widely because of its unique ORR electrocatalytic activity [4]. However, the high cost, poor durability and the possibility of poisoning limit the development of Pt catalysts [5–7]. To find alternative electrocatalysts, scientists have evaluated many materials, including transition metal oxides [8], nitrides [9], and sulfides [10] in recent years, but their electrocatalytic activity is still not comparable to that of Pt. Therefore, reducing the content of Pt while ensuring high catalytic performance has become the key to solving the problem.

Generally, introducing other metals, preparing nanoparticles with special morphology and finding suitable carriers are the methods to reduce the amount of platinum [11–14]. The synergy between Pt and additional metals can effectively improve the activity and stability of electrocatalyst, which has attracted extensive exploration [15–17]. The research shows due to the mismatch of lattice, the introduction of transition metal will lead to the shrinkage of Pt crystal and the downward movement of the d-band center, which will change the binding strength of Pt-oxygen intermediates and improve the catalytic activity of ORR [18]. Among many metals, Pd, with a similar lattice constant and crystal structure to Pt, may be an attractive candidate. As a member of the Pt group elements, Pd is much cheaper and relatively abundant than Pt [19]. In addition, the ORR activity of Pd is just slightly lower than Pt and shows the utmost resistance to carbon monoxide poisoning [20,21]. Intuitively, the introduction of Pd should not only improve the ORR activity but also improve the stability of the electrocatalyst. Chen et al., prepared PtPd/CNWs nanowire by the centrifugal electrospinning method [22]. The one-dimensional nanostructured catalysts showed excellent ORR activity. the half-wave potential of PtPd/CNWs (0.865 V) shows a positive shift of about 52 mV relative to that of 20% Pt/C (0.813 V). After 5000 potential cycles, the electrochemically active surface area of PtPd/C nanowire almost did not change, while Pt/C decreased by nearly 40%. Liu et al. developed ordered mesoporous carbons that supported PtPd nanoclusters towards ORR with a greater tolerance to methanol [23].

After the addition of methanol, only a 5.8% decrease of the current was observed for $Pt_3Pd_1NCs/OMC$, much lower than that of Pt/C (39.2%), which can be mainly attributed to the Pd around the Pt inhibiting the adsorption of methanol on the active center.

Transition metals such as Co and Fe are widely used as ORR catalysts because of their excellent electrochemical performance and low cost [24–26]. A series of studies have shown that the coordination interaction between Co and N-rich carbon can form effective ORR activity sites, such as carbon-coated metallic nanoparticles (Co@C), a Co nitrogen structure ($Co-N_2$, $Co-N_4$), and carbon-coated Co carbides ($Co_xC_y$@C) [27]. Another important factor determining the performance of oxygen reduction is the well-designed structure. Zeolitic imidazolate frameworks (ZIFs) are a type of porous material with good structure and a large specific surface area, which can promote rapid material transport and accessible active sites [28]. Meanwhile, ZIF-67 derived materials usually produce Co-N-C species. Many studies have shown that the defective nanocarbons containing Co-N-C have abundant active sites, which is beneficial to improving the catalytic activity and accelerating the charge transfer in ORR [29–31]. Based on these noteworthy advantages, ZIF-67 has become a promising template for the preparation of Co-N-C structure by simple thermal decomposition. However, the ORR activity of pure Co-N-C catalyst is hardly comparable with commercial Pt/C, and its stability under acidic conditions is difficult to guarantee. Based on the above factors, we supported Pt/Pd nanoparticles on ZIF-67 to prepare high-performance ORR catalysts. The existence of Co will reduce the OH adsorption on Pt, leading to an increase in catalytic activity [17]. In addition, the introduction of Co can also improve durability by reducing the dissolution and migration of Pt in the ORR process [32]. On the other hand, the imidazole group in the precursor of ZIF-67 can provide rich nitrogen atoms, which makes it easy to obtain N-doped carbon materials. Many studies have shown that doping nitrogen into the carbon skeleton can change the local charge density distribution of bonded carbon atoms and produce N-rich active sites, so as to promote the adsorption of oxygen on the electrocatalyst [33,34]. The synergistic effect between N-doped carbon and metal nanoparticles can increase the electron density of catalytic sites and further improve ORR activity, which makes it a popular trend to load metals nanoparticles with N-doped carbon. For instance, Han et al. [35] prepared PtZn intermetallic nanoparticles anchored on conductive NC carriers by pyrolysis ZIF-8 graphene oxide composites with coordinated Pt ions by tetra(4-carboxyphenyl)porphine, which significantly enhanced oxygen reduction activity and stability. The half-wave potential of PtZn/NC composites (0.911 V) exhibits a positive shift about 36 mV relative to that of Pt/C (0.875 V) and almost no change was demonstrated after 5000 potential cycles.

In this work, we loaded the prepared Pt and Pd nanoparticles on ZIF-67 and carbonized them to form a nitrogen-doped carbon material loaded with Pt, Pd, and Co, while effectively reducing the content of Pt, Pt/Pd/Co-N-C also shows excellent ORR activity and stability. This can mainly be attributed to the synergy between Pt, Pd, and Co, as well as the contribution of nitrogen-doped carbon carriers.

## 2. Result and Discussion

The structure and morphology of Pt/Pd/ZIF-67 and Pt/Pd/Co-N-C were characterized by SEM and TEM. As shown, the Pt/Pd/ZIF-67 (Figure 1A,B) has a rhombododeca-hedron shape and an average particle size of about 500 nm, which is the typical structure of ZIF-67 [36,37]. After pyrolysis at 700 °C, the Pt/Pd/Co-N-C still retains the polyhe-dron mechanism and shows the shape of a rough ridge. At the same time, the size of the Co-N-C frame is reduced by dozens of nanometers compared with the ZIF-67 without pyrolysis, which is consistent with other reports [25]. The results can be attributed to the rapid skeleton shrinkage associated with ligand carbonization and atomic migration during pyrolysis [38]. In addition, no metal nanoparticles were observed on the surface under this magnification. TEM analysis further reveals the morphological information of the Pt/Pd/Co-N-C catalyst in Figure 1C,D). As shown, there are some black dots with the lattice fringe of 0.198 nm and 0.137nm, which can be indexed to the {200} and {220} planes

of PtPd alloy, respectively [39–41], indicating that Pt and Pd may form alloy structure. In addition, the results show that the average size of Pt/Pd on the surface of the Co-N-C frame is about 2.4 nm, which may possess a high electrocatalytic activity in terms of particle size effect. Furthermore, the EDS images of Pt/Pd/Co-N-C further suggest the uniform distribution of N, O, Co, Pt, and Pd elements on the carbon skeleton (Figure S1), evidencing the successful combination of Pt/Pd alloy and Co nanoparticles with N-doped carbon carrier.

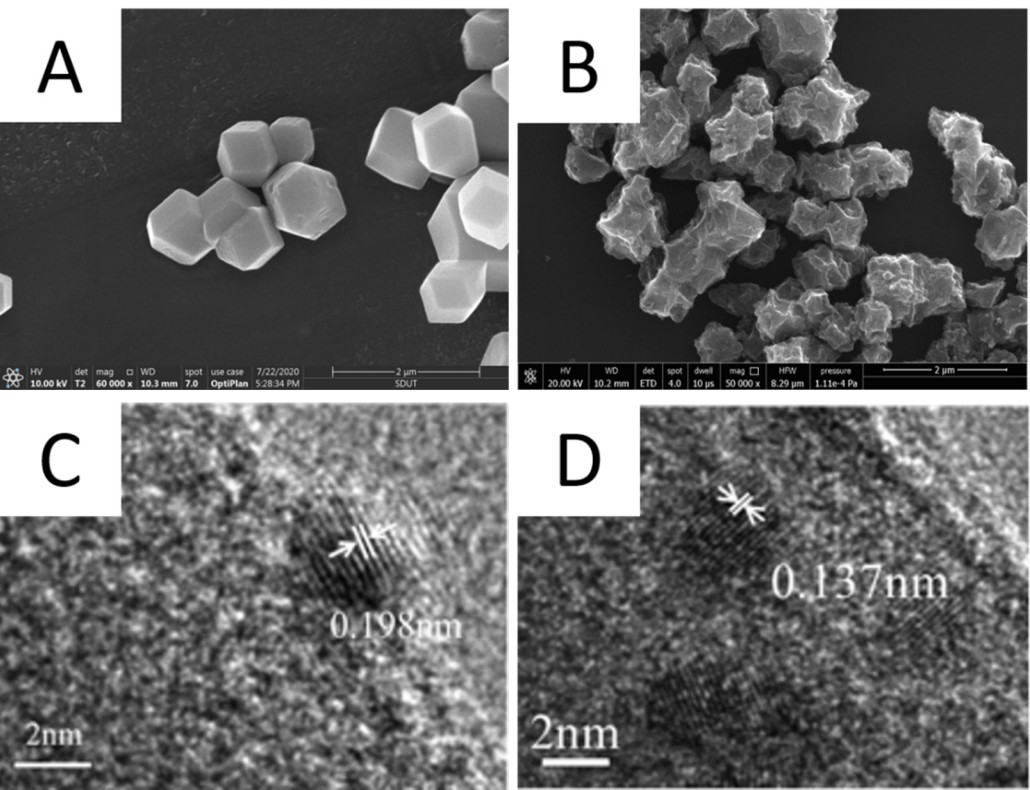

**Figure 1.** SEM images of Pt/Pd/ZIF-67 (**A**), and Pt/Pd/Co-N-C (**B**); HRTEM image of Pt/Pd/Co-N-C (**C**,**D**).

The Raman spectra of Pt/Pd/Co-N-C are shown in Figure 2. As shown, typical disordered carbon (D-band) and graphite carbon (G-band) structures were observed at 1356 cm$^{-1}$ and 1590 cm$^{-1}$. The $I_D/I_G$ of Pt/Pd/Co-N-C is 1.17. The high $I_D/I_G$ can be attributed to the introduction of topological disorder into the graphite layer. Where the bonding is still mainly sp$^2$, but the weaker bonds soften the vibrational modes [42–45]. These defect sites can optimize the adsorption energy of the catalytic step by adjusting the electronic and surface properties of the electrocatalyst [46].

To further investigate the elemental composition of Pt/Pd/Co-N-C, the samples were characterized by XPS. It is found from the full spectrum of Pt/Pd/Co-N-C (Figure S2) that the catalyst contains platinum, palladium, cobalt, nitrogen, and oxygen elements. Table S1 shows the element contents of Pt/Pd/Co-N-C. In the N1s high-resolution spectra of Pt/Pd/Co-N-C (Figure 3a), the peaks at 398.8 eV, 400.3 eV, 403.1 eV, and 406 eV can be corresponded to pyridinic-N, pyrrolic-N/Co-Nx, graphitic nitrogen, and oxidized-N, respectively [47–50], where pyridinic-N and pyrrolic-N/Co-Nx are the main types. It is reported that N-containing MOF-derived catalysts usually result in Co-N-C and Co-N species; both of them are proposed to be active toward ORR [51]. Moreover, importantly, pyridinic-N can improve the catalytic activity of electrode materials because it can enhance the electron density and doping properties [34,52]. Pyrrolic-N usually shows fast charge mobility and good charge transfer between the donor and the acceptor and is considered to improve the catalytic activity by reducing the carbon band gap energy [53]. In addition, the presence of Graphite-N also has a positive effect on ORR activity, it can significantly

improve the limiting current density. In the Co2p high-resolution spectra of Pt/Pd/Co-N-C (Figure 3b), the peaks at 779.5 and 794.3 eV were assigned to metallic Co(0). The peaks around 780.8 and 797.1 eV correspond to $Co2p_{3/2}$ and $Co2p_{1/2}$. The presence of $Co-N_x$ can be demonstrated by the peak at 782.1 eV [54–56]. Co-Nx is usually regarded as the catalytic activity center of ORR, especially $Co-N_4$, which may not only contribute to the overall ORR activity but also improve the durability and service life of the electrocatalyst [16]. In the Pt4f high-resolution spectra of Pt/Pd/Co-N-C (Figure 3c), the peaks at 71.8 and 75.3 eV correspond to $Pt4f_{7/2}$ and $Pt4f_{5/2}$, respectively [57]. Meanwhile, Pt4f can be divided into two pairs of peaks. The peaks at 71.8 eV and 75.1 eV are the characteristic peaks of Pt(0), while the peaks at 72.8 eV and 76.4 eV can be attributed to Pt(II). The results show that Pt atoms mainly exist in the metal form of Pt(0), which is beneficial for the enrichment of catalytic activity [18]. In the Pt4f high-resolution spectra of Pt/Pd/Co-N-C (Figure 3d), the peaks at 335 eV and 340.5 eV are the characteristic peaks of Pd(0), while the peaks at 338.2 eV and 343.3 eV can be attributed to Pd(II) [58]. The results show that Pd atoms mainly exist in Pd(0) and Pd(II) states. In addition, the Pt4f and Pd3d peaks of Pt/Pd/Co-N-C show an obvious shift compared to those in Pt/C and Pd/C [59], suggesting that the electronic structures of Pt and Pd have changed, and thus suggesting that the electronic structures of Pt have changed [59], which can mainly be due to the electronic ensemble effect of Pt and Pd atoms [11]. The change of binding energy between Pt and Pd further indicates that an alloy structure may be formed between them.

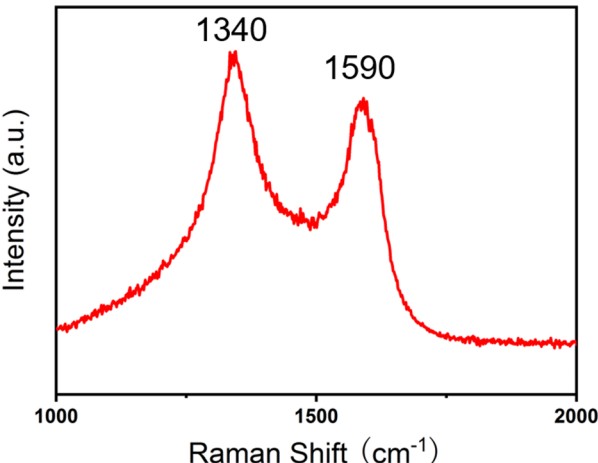

**Figure 2.** Raman spectra of Pt/Pd/Co-N-C.

To evaluate the ORR catalytic activity, the linear sweep voltammetry (LSV) test was performed on Pt/Pd/Co-N-C and other catalysts (Figure 4A) in a 0.1 M KOH solution. As shown, the half-wave potential ($E_{1/2}$) and current density of Pt/Pd/Co-N-C (0.84 V and 6.6 mA cm$^{-2}$) are the most positive alternatives to those of Co-N-C (0.82 V and 4.4 mA cm$^{-2}$), Pd/Co-N-C (0.81 V and 5.4 mA cm$^{-2}$), Pt/Co-N-C (0.81 V and 5.9 mA cm$^{-2}$), and Pt/C (0.84V and 5.8 mA cm$^{-2}$) under the same experimental conditions. Compared with Co-N-C, the current densities of Pt/Co-N-C and Pd/Co-N-C are increased significantly, which is due to the introduction of Pt or Pd nanoparticles, which is beneficial to increasing the electroactive area [60]. In addition, Pt/Pd/Co-N-C showed better catalytic performance, indicating a good synergy between Pt and Pd. From the change in Pt4f binding energy in XPS, it can also be inferred that its electronic state has changed, which may be the reason for the enhancement of ORR. To further study the ORR kinetics, the electron transfer number (n) at different spinning speeds was calculated by the RDE experiment. With the increase in rotating speed, the current density of Pt/Pd/Co-N-C increases significantly, while the initial potential remains constant (Figure 4B). According to the K–L plots (Figure 4C), the as-calculated (n) values of Pt/Pd/Co-N-C are 3.8, indicating a good four-electron selectivity. In other words, almost no intermediate products are produced in

the process of ORR, which can provide higher energy efficiency in the application of fuel cells. As shown in Figure 4D, The Tafel slope of Pt/Pd/Co-N-C (74 mV dec$^{-1}$) is very close to that of Pt/C (72 mV dec$^{-1}$), which means a fast ORR process. The ORR performance of all prepared samples and the comparison with other catalysts are shown in Table S2. Considering the low noble metal loading, Pt/Pd/Co-N-C still has excellent performance comparable to Pt/C catalyst, which may be a potential ORR catalyst.

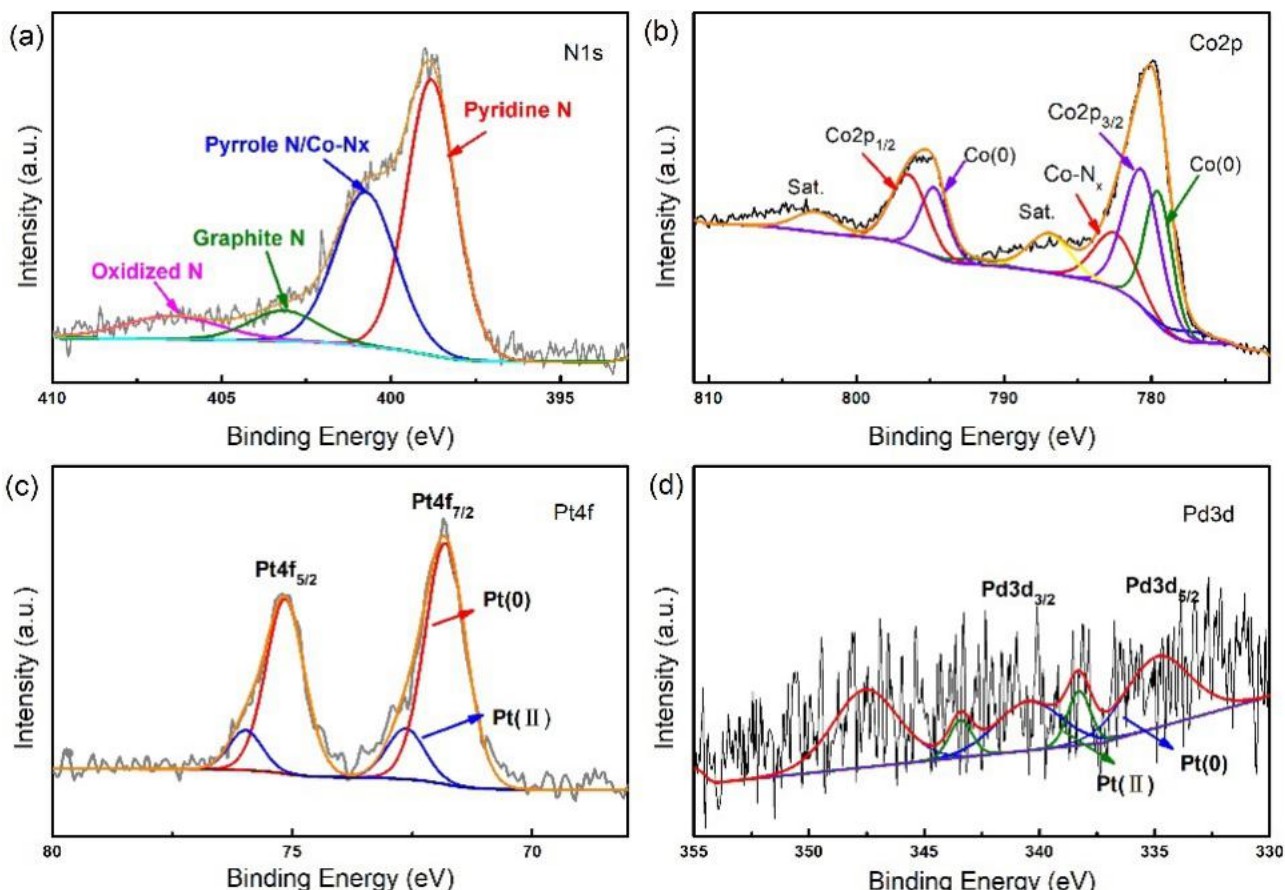

**Figure 3.** N1s XPS spectra (**a**), Co2p XPS spectra (**b**), Pt4f XPS spectra (**c**), and Pd3d XPS spectra(**d**) of Pt/Pd/Co-N-C.

In addition to the required high activity, stability is also an important index to evaluate the performance of catalysts. Figure 5A shows the chronoamperometric responses to the addition of 1 M methanol for Pt/Pd/Co-N-C and Pt/C. As shown, the current density of Pt/Pd/Co-N-C and Pt/C changed significantly after the addition of methanol. When tested to 1000 s, the current of Pt/Pd/Co-N-C remains at 91% compared with the original value—much higher than that of Pt/C (59%)—indicating that the methanol resistance of Pt/Pd/Co-N-C is much higher than that of commercial Pt/C. Compared with the unprotected PtPd nanoclusters, Pt/Pd/Co-N-C displays less current change after adding methanol, which may be due to the protective effect of the N-doped carbon carrier [23,25]. Figure 5B shows the durability test of Pt/Pd/Co-N-C and Pt/C. The current density of Pt/Pd/Co-N-C and Pt/C decreased slowly on the whole. After the 36,000 s test, the current density of Pt/Pd/Co-N-C decreased to 90% of the initial value, while Pt/C decreased to 68%. These results show that the stability of Pt/Pd/Co-N-C is much better than that of Pt/C. The introduction of Pd and the contribution of N-doped carbon support jointly, promote the high stability of the catalyst.

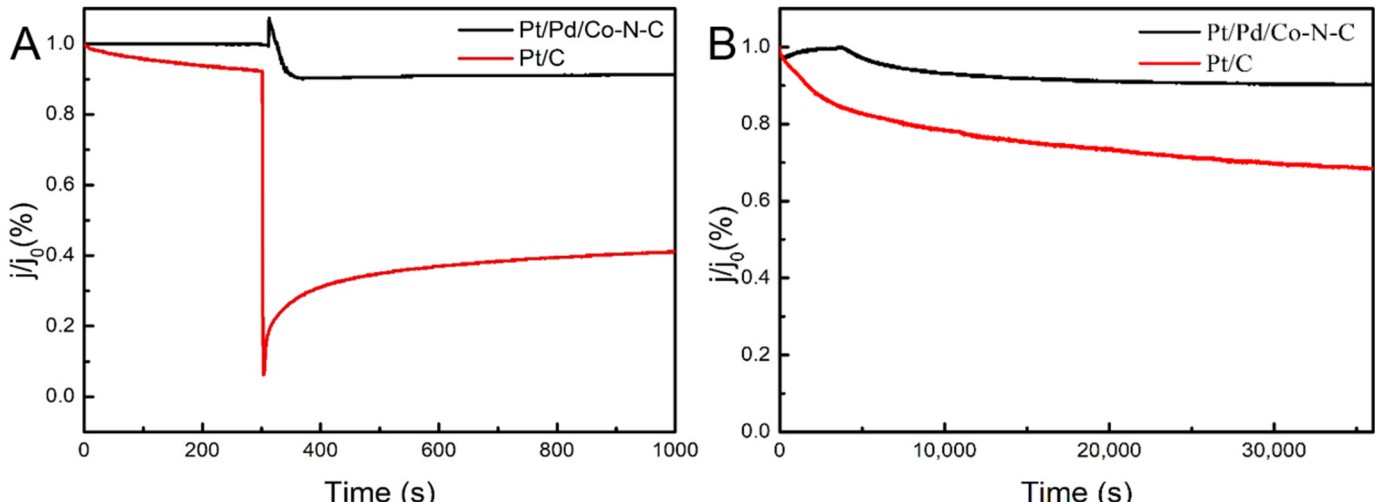

**Figure 4.** The LSV curve of several catalysts in 0.1M KOH saturated with oxygen (**A**); The LSV curve of Pt/Pd/Co-N-C at different speeds in 0.1 M KOH saturated with oxygen (**B**); K–L curve and electron transfer number of Pt/Pd/Co-N-C (**C**); Tafel slopes of several materials (**D**).

**Figure 5.** Chronoamperometric responses to the addition of 1 M methanol for Pt/Pd/Co-N-C and Pt/C (**A**); Chronoamperometric responses of Pt/Pd/Co-N-C and Pt/C with a rotating speed of 1600 rpm in $O_2$-saturated 0.1 M KOH (**B**).

## 3. Experimental

### 3.1. Reagents and Apparatus

Potassium hexachloroplatinate ($K_2PtCl_6$), Palladium chloride ($PdCl_2$), and polyvinylpyrrolidone (PVP) were purchased from Macklin Biochemical Co., Ltd. (Shanghai, China). Cobalt nitrate hexahydrate ($Co(NO_3)_2 \cdot 6H_2O$) was purchased from Fuchen Chemical Reagent Co., Ltd. (Tianjin, China). 2-methylimidazole (2-MEIM) was purchased from Macklin Biochemical Co., Ltd. (Shanghai, China). Nafion solution (5 wt%) was purchased from Jingzhong Co., Ltd. (Shanghai, China). Deionized water (18.25 M$\Omega$ cm$^{-1}$) was used to prepare all aqueous solutions throughout the experiments. All reagents were used without further purification.

Scanning electron microscopy (SEM, Oberkochen, Germany) analyses were recorded with an FEI SIRION microscope. Transmission electron microscopy (TEM) images and energy-dispersive spectrometry (EDS) analysis were obtained using an FEI Tecnai G2T20 at 200 kV. Raman spectrum was taken at 532 nm and recorded on the LabRAM HR800 from JY Horiba (Kowloon, Hong Kong, China). X-ray photoelectron spectroscopy (XPS) was performed on a K-alpha (Thermo Scientific, Waltham, MA, USA) using the C1s peak (284.8 eV) as the reference for calibration. The electrochemical characterizations, including cyclic voltammetry (CV), chronoamperometry, linear sweep voltammetry (LSV), and chronoamperometry (CA) were performed with a CS2355 electrochemical workstation (CorrTest Instrument, Wuhan, China) and rotating disk electrode (RDE) (AFMSRX, PINE Instruments, Grove City, PA, USA). A three-electrode system was used in all experiments with the working electrode as a glassy carbon electrode (GCE) (5 mm diameter, 0.196 cm$^2$ area), respectively, Hg/HgO electrode and a platinum plate electrode, applied as counter and reference electrodes using a rotating disk electrode (RDE).

### 3.2. Material Syntheses

#### 3.2.1. Preparation of Pt/Pd Nanoparticles

In a typical synthesis, 39 mg of $K_2PtCl_6$ and 14 mg of $PdCl_2$ were dissolved in 10 mL of deionized water, respectively, and mixed evenly for standby. An amount of 333 mg of PVP was dissolved in 100 mL of methanol and transferred to the flask for heating in a water bath cauldron at 80 °C. Then, the prepared $K_2PtCl_6$ and $PdCl_2$ solutions were quickly added to the flask with agitation and maintained for 3 h. Finally, methanol was removed by rotary evaporator, and PtPd nanoparticles were obtained and dispersed in water for standby. Similarly, only 39 mg of $K_2PtCl_6$ or 14 mg of $PdCl_2$ were added to the reaction system to prepare Pt nanoparticles and Pd nanoparticles.

#### 3.2.2. Preparation of ZIF-67

Typically, 1.45 g of $Co(NO_3)_2 \cdot 6H_2O$ and 3.28 g of 2-methylimidazole were dissolved in methanol and stirred for 1h to mix evenly. The mixed solution was allowed to stand at room temperature for 24 h and then centrifuged to collect the solid. The obtained product was dried in a vacuum oven at 60 °C for 12 h. Finally, ZIF-67 powder was obtained by grinding.

#### 3.2.3. Preparation Pt/Pd/Co-N-C Composite

Pt/Pd/ZIF-67 were synthesized via an impregnation method. The prepared Pt/Pd nanoparticles were added into ZIF-67 powder methanol solution and stirred at room temperature for 4 h to mix evenly. The mixed solution was allowed to stand at room temperature for 12 h. Then, the obtained composite was washed several times with deionized water and ethanol, dried, and labeled as Pt/Pd/ZIF-67.

Finally, the prepared Pt/Pd/ZIF-67 was pyrolyzed under an Ar$_2$ atmosphere at 700 °C for 2 h with a heating rate of 5 °C min$^{-1}$ and labeled as Pt/Pd/Co-N-C. For comparison, Pt/Co-N-C, Pd/Co-N-C, and Co-N-C were also prepared.

### 3.3. Electrocatalytic Measurements

The catalyst ink was prepared by a mixture of 4.0 mg of catalyst in 1 mL of solution containing 700 μL of deionized water, 250 μL of isopropanol, and 50 μL of Nafion (5 wt%). After ultrasonic treatment for 1 h, 10 μL catalyst ink was loaded onto an RDE and used as a working electrode for electrochemical tests after natural drying. The catalyst loading for the RDE is 0.2 mg cm$^{-2}$. The electrolyte used was 0.1 M KOH, and the test temperature was 30 °C. In the ORR test, the electrolyte is $O_2$ saturated.

The Koutecky–Levich equation was used to calculate the kinetic current, which can be described as follows:

$$\frac{1}{j} = \frac{1}{j_k} + \frac{1}{j_d} = \frac{1}{j_k} + \frac{1}{B\omega^{1/2}} \tag{1}$$

where $j$ is the measured current density, $j_k$ and $j_d$ are the kinetic and diffusion-limited current densities, respectively. $\omega$ is the electrode rotating rate, and $B$ could be determined from the slope of the K–L plots based on the Levich equation as follows:

$$B = 0.62nF(D_o)^{2/3}\vartheta^{-1/6}C_o \tag{2}$$

where n represents the electron transfer number, F is the Faraday constant, $D_o$ is the diffusion coefficient of $O_2$ in 0.1 M KOH, $\vartheta$ is the kinetic viscosity and $C_o$ is the bulk concentration of $O_2$.

Tafel slopes were calculated using the Tafel equation:

$$\eta = a + blg \mid j_k \mid \tag{3}$$

where η is the overpotential, $j$ is the disk current density, and b is the Tafel slope.

## 4. Conclusions

A simple and efficient method was developed to prepare N-doped carbon materials loaded with Pt, Pd, and Co, intended for the ORR. The content of total precious metals in Pt/Pd/Co-N-C is only 0.56%, and the ORR activity is comparable to that of Pt/C. In addition, Pt/Pd/Co-N-C showed much higher stability than Pt/C. Overall, while ensuring the performance of ORR, we reduced the content of Pt by introducing other metals and designing appropriate supports, which provides a new strategy for the development of low-cost and high activity ORR electrocatalysts (References [61–64] are cited in the Supplementary Materials).

**Supplementary Materials:** The following are available online at https://www.mdpi.com/article/10.3390/catal12050482/s1, Figure S1. TEM of Pt/Pd/Co-N-C(A) and EDS mapping of N, O, Co, Pt, and Pd(B–F); Figure S2. XPS surveys of Pt/Pd/Co-N-C; Figure S3. CVs on Pt/Pd/Co-N-C in $N_2$ and $O_2$ saturated 0.1 M KOH solution; Table S1. Atomic content of Pt/Pd/Co-N-C; Table S2. ORR activity data from different catalysts.

**Author Contributions:** Conceptualization, Methodology, Investigation, Visualization, Data Curation, Writing—Original Draft, Y.J.; Visualization, Formal analysis, D.Z., X.Z., Z.C. and L.Z.; Methodology, Visualization, Investigation, Writing—review & editing, Y.C.; Supervision, Writing—review & editing, Resources, Funding acquisition, W.S. All authors have read and agreed to the published version of the manuscript.

**Funding:** This work was supported by the National Natural Science Foundation of China 278 (51502161), the Natural Science Foundation of Shandong Province (ZR2020ME041), Joint Zibo-SDUT Fund (2019ZBXC358), and the Foundation of State Key Laboratory of Biobased Material and Green Papermaking, Qilu University of Technology, Shandong Academy of Sciences (KF2019-06).

**Data Availability Statement:** The data presented in this study are available on request from the corresponding author.

**Conflicts of Interest:** The authors declare no conflict of interest.

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
