# Peer review of "Pt/Pd Decorate MOFs Derived Co-N-C Materials as High-Performance Catalysts for Oxygen Reduction Reaction"

_catalysts, doi:10.3390/catal12050482_

Round 1
Reviewer 1 Report
The article presents a study on the preparation and activity evaluation of the composite catalysts for the ORR reaction based on ZIF-67.
Unfortunately, there are many shortcomings in the work, such as:
In the work, the authors refer to supporting materials, which, unfortunately, are not presented. This leads to difficulties in reading the article.
In the literature review, the choice of an alkaline medium for studying the catalysts activity is not justified, while most fuel cells use a proton-exchange membrane and, accordingly, it is necessary to evaluate the catalysts activity in an acidic medium.
In the study, there are not enough methods for characterizing the structure, it is necessary to add XRD. So, it is not clear what form PtPd is in the PtPd/Co-N-C material - two separate phases, a solid solution, or a shell-core structure. The title of the article indicates Pt/Pd, and further in the text PtPd, it is not clear what this means.
The materials composition must be determined by elemental analysis or TEM EDAX to determine the Pt-Pd atomic ratio.
What is the mass fraction of metals after heat treatment in materials? According to the XPS data, the signal intensity from Pd is extremely low (strong noise). What is the content of palladium in the material, is it correct to interpret the spectrum with such a weak signal and strong noise?
In the experimental part, it is necessary to indicate the catalyst loading at RDE in g/cm2.
Why in Figure 4A the value of the limiting diffusion current for different materials varies significantly, while the electrode area and the electrolyte composition are the same.
The experimental part and the experimental part do not indicate at what potential the chronoamperometry was used, what is the reason for the choice of this potential?
Figure 1cd presents the TEM data, why only a few nanoparticles are sized, it is necessary to size at least 100 nanoparticles for each material, present a histogram of the nanoparticle size distribution, and determine the average nanoparticle size for each material.
typo K2Pt/Cl6 in several places throughout the article
typo page 3 3 - Ar2
Reviewer 2 Report
The study presents a novel catalyst formulation for ORR to reduce precious metal usage in electrochemical systems. The catalyst concept is novel and interesting, but the study has significant omissions in their reporting which makes it impossible to assess the advances made by this catalyst. Limiting currents are a reflection of active area accessibility and not catalytic activity. Likewise, the study does not compare their results to the widely studied Pt/C results from other researchers for comparable loadings and active areas (both of which do not appear to be quantified). Until these values are reported, and the control data is confirmed to be reliable, the manuscript is not of sufficient quality to be reported.
Introduction
- Most of the comments in the introduction about improvements to stability and performance are ambiguous with no quantification. These claims should be quantified relative to standard values reported by previous studies to clearly set the metrics by which the proposed study will be assessed.
- Page 2: “ORR catalysts because of their excellent electrochemical and low cost.” Electrochemical what?
Experimental Section
- This section should state that the study uses a rotating disc electrode.
- Section 2.2.1. What heating temperature and apparatus was used?
- Section 2.3. Do you confirm how much catalyst was applied to the working electrode?
- What was the test solution, temperature and oxygen partial pressure used for the electrochemical characterization?
Results
- The statements about the Id/IG ratio are misleading. If the Id/Ig ratio is very low, it can be either amorphous carbon or highly oriented graphite. With high disorder, the Id/Ig ratio will decrease not increase. See the follow reference about amorphization trajectories for carbon-based materials via Raman. Ferrari, A. C. & Robertson, J. Raman spectroscopy of amorphous, nanostructured, diamond-like carbon, and nanodiamond. Trans. R. Soc. A Math. Phys. Eng. Sci. 362, 2477–2512 (2004).
- How many tests were run? No error or standard deviations are provided. Any evidence that these results are statistically significant?
- Were the active areas and loadings of each catalyst material measured? If so, what were these values?
- These results should be compared to previous studies with ORR to confirm the reliability of the current tests. Exchange current densities, Tafel Slopes and limiting currents for the Pt/C catalysts have been measured many times. How do the values presented here compare?
Reviewer 3 Report
The authors reported the synthesis of PtPd/CoNC composites and studied the ORR activity in 0.1 M KOH. There are some issues that the authors need to address before the paper is published
- TEM measurements show the formation of Pt and Pd NPs. These are separated. How would they show any "synergistic" interactions in ORR electrocatalysis? In fact, no interaction between Pt and Pd is detected in XPS measurements either. The authors need to examine their data more carefully and see if there is any PtPd alloy NPs and how they interact with each other.
- What are the metal contents in the samples by XPS, ICP, etc?
- CVs of the samples need to be included, at least in the SI, so that the butterfly features can be used to calculate the electrochemical surface area and the ORR specific activity. Also the authors may want to compare the mass activity among the samples.
- There is no clear contribution of CoNC to the ORR activity. What is the benefit of having CoNC in the system? Why not just use plain carbon?
Round 2
Reviewer 1 Report
The authors responded to the comments and made the necessary changes to the text of the article. The article may be accepted for publication.
Author Response
Many thanks for reviewer!
Reviewer 2 Report
Comment 1: How many tests were run? No error or standard deviations are provided. Any evidence that these results are statistically significant?
Reply 2: While I understand that you may not be able to run more tests, you need to report how many were run. Repeatability is a critical metric that should be demonstrated to ensure that the difference between measurements is not random error. So, the number of runs and some error estimations are needed before publication as many of the results reported are within very narrow potential and current windows.
Comment 2: These results should be compared to previous studies with ORR to confirm the reliability of the current tests. Exchange current densities, Tafel Slopes and limiting currents for the Pt/C catalysts have been measured many times. How do the values presented here compare?
Reply 2: Again, you should report the exchange current densities and Tafel slopes of all the materials that you examined and compare them to literature values. These are better metrics than half-wave and onset potentials. Figure 4 shows that this data was collected but you do not analyze or report it. This should be done before publication.
Comment 3: The authors report the loading but they do not specify how this was checked. How did you confirm you reached the desired loading amounts on your samples?
Reviewer 3 Report
The authors addressed some of the issues raised previously. However, problems remain.
- The authors ascribed the NPs in Fig 1c and 1d to separate Pt and Pd nanoparticles. Yet prior research has shown that PtPd alloy NPs also show similar lattice fringes. The authors need to look more carefully into the possibility of the formation of PtPd alloy NPs in the present study, in particular in light of the elemental mapping results in Fig S1. WIth the formation of PtPd alloy NPs, it is more likely to see any synergistic interactions between Pt and Pd.
- The statement of "The negative shift of Pt indicates the lowered d-band centers, which can enhance the surface catalytic activity[56]" is incorrect. A lower BE means electron-richer Pt, and hence a upshift of the d band center and a stronger interaction with oxygen intermediates. This will eventually leads to a lower ORR activity.
- This statement is confusing, "In addition to the synergistic effect of Pt and Pd, the excellent ORR performance can also be attributed to the rich nitrogen atoms and Co-N-C active centers in the carbon ma-terial carrier derived from ZIF-67, which can effectively enhance the interaction between Pt nanoparticles and carbon surface[30]." Did the authors carry out control experiments with PtPd deposited on regular carbon black and compare the ORR activity? There are too many components in the system. This makes it difficult to pinpoint the exact ORR active sites.
Round 3
Reviewer 2 Report
1). The number of repeat tests and error need to be reported in the manuscript. Your results are very close to each other so this is needed for accurate interpretation.
2). Simple mass gain measurements can be used to confirm your catalyst loadings, this should be reported. The electrochemical active surface area (ECSA) should also be reported. Without these parameters it is impossible to understand any improvements made by these catalyst materials.
3). The K-L data was collected over a huge potential range but the region with kinetic influence is quite small. How did you obtain kinetic data from the limiting current region, where the response is independent of potential?
4). The Pt/C values (the control test) were not compared with literature data for exchange current density and Tafel slopes. This should be added.
Reviewer 3 Report
revision is good.